# Learning Latent Permutations with Gumbel-Sinkhorn Networks

**Gonzalo E. Mena** [*]
Department of Statistics,
Columbia University
gem2131@columbia.edu

**David Belanger**
Google Brain

**Scott Linderman**
Department of Statistics,
Columbia University

**Jasper Snoek**
Google Brain

## Abstract

Permutations and matchings are core building blocks in a variety of latent variable models, as they allow us to align, canonicalize, and sort data. Learning in such models is difficult, however, because exact marginalization over these combinatorial objects is intractable. In response, this paper introduces a collection of new methods for end-to-end learning in such models that approximate discrete maximum-weight matching using the continuous Sinkhorn operator. Sinkhorn operator is attractive because it functions as a simple, easy-to-implement analog of the softmax operator. With this, we can define the Gumbel-Sinkhorn method, an extension of the Gumbel-Softmax method (Jang et al., 2016; Maddison et al., 2016) to distributions over latent matchings. We demonstrate the effectiveness of our method by outperforming competitive baselines on a range of qualitatively different tasks: sorting numbers, solving jigsaw puzzles, and identifying neural signals in worms.

## 1 Introduction

In principle, deep networks can learn arbitrarily sophisticated mappings from inputs to outputs. However, in practice we must encode specific inductive biases in order to learn accurate models from limit data. In a variety of recent research efforts, practitioners have provided models with the ability to explicitly manipulate latent combinatorial objects such as stacks (Dyer et al., 2015; Joulin & Mikolov, 2015), memory slots (Graves et al., 2014; Sukhbaatar et al., 2015), mathematical expressions (Neelakantan et al., 2015), program traces (Gaunt et al., 2016; Bošnjak et al., 2017), and first order logic (Rocktäschel & Riedel, 2017). Operations on these discrete objects can be approximated using differentiable operations on continuous relaxations of the objects. As such, these operations can be included as modules in neural network models that can be trained end-to-end by gradient descent.

Matchings and permutations are a fundamental building block in a variety of applications, as they can be used to align, canonicalize, and sort data. Prior work has developed learning algorithms for supervised learning where the training data includes annotated matchings (Caetano et al., 2009; Petterson et al., 2009; Tang et al., 2016). However, we would like to learn models with latent matchings, where the matching is not provided to us as supervision. This is a common and relevant setting. For example, Linderman et al. (2017) showed a problem from neuroscience involving the identification of neurons from the worm C. elegans can be cast as the inference of latent permutation on a larger hierarchical structure.

Unfortunately, maximizing the marginal likelihood for problems with latent matchings is very challenging. Unlike for problems with categorical latent variables, we cannot obtain unbiased stochastic gradients of the marginal likelihood using the score function estimator (Williams, 1992), as computing the probability of a given matching requires computing an intractable partition function for a structured distribution. Instead, we draw on recent work that obtains biased stochastic gradients by relaxing the discrete latent variables into continuous random variables that support the reparametrization trick (Jang et al., 2016; Maddison et al., 2016).

---

[*]Work done while the author was at Google Brain.

Our contributions are the following: first, in Section 2 we present a theoretical result showing that the non-differentiable parameterization of a permutation can be approximated in terms of a differentiable relaxation, the so-called *Sinkhorn operator*. Based on this result, in Section 3 we introduce *Sinkhorn networks*, which generalize the work of method of Adams & Zemel (2011) for predicting rankings, and complements the concurrent work by Cruz et al. (2017), by focusing on more fundamental aspects. Further, in Section 4 we introduce the *Gumbel-Sinkhorn*, an analog of the Gumbel Softmax distribution (Jang et al., 2016; Maddison et al., 2016) for permutations. This enables optimization of the marginal likelihood by the reparametrization trick. Finally, in Section 5 we demonstrate that our methods outperform strong neural network baselines on the tasks of sorting numbers, solving jigsaw puzzles, and identifying neural signals from C. elegans worms.

## 2 THE SINKHORN OPERATOR: AN ANALOG OF THE SOFTMAX FOR PERMUTATIONS

One sensible way to approximate a discrete category by continuous values is by using a temperature-dependent softmax function, component-wise defined as $\text{softmax}_\tau(x)_i = \exp(x_i/\tau)/\sum_{j=1} \exp(x_j/\tau)$. For positive values of $\tau$, $\text{softmax}_\tau(x)_i$ is a point in the probability simplex. Also, in the limit $\tau \to 0$, $\text{softmax}_\tau(x)_i$ converges to a vertex of the simplex, a one-hot vector corresponding to the largest $x_i$ [1]. This approximation is a key ingredient in the successful implementations by Jang et al. (2016); Maddison et al. (2016), and here we extend it to permutations.

To do so, we first state an analog of the normalization implemented by the softmax. This is achieved through the Sinkhorn operator (or Sinkhorn normalization, or Sinkhorn balancing), which iteratively normalizes rows and columns of a matrix. Specifically, following Adams & Zemel (2011), we define the Sinkhorn operator $S(X)$ over an $N$ dimensional square matrix $X$ as:

$$
\begin{aligned}
S^0(X) &= \exp(X), \\
S^l(X) &= \mathcal{T}_c\left(\mathcal{T}_r(S^{l-1}(X))\right), \\
S(X) &= \lim_{l \to \infty} S^l(X).
\end{aligned}
\tag{1}
$$

where $\mathcal{T}_r(X) = X \oslash (X\mathbf{1}_N\mathbf{1}_N^\top)$, and $\mathcal{T}_c(X) = X \oslash (\mathbf{1}_N\mathbf{1}_N^\top X)$ as the row and column-wise normalization operators of a matrix, with $\oslash$ denoting the element-wise division and $\mathbf{1}_N$ a column vector of ones. Sinkhorn (1964) proved that $S(X)$ must belong to the Birkhoff polytope, the set of doubly stochastic matrices, that we denote $\mathcal{B}_N$ [2].

Building on our analogy with categories, notice that choosing a category can always be cast as a maximization problem: the choice $\arg\max_i x_i$ is the one that maximizes the function $\langle x, v \rangle$ (with $v$ being a one-hot vector), i.e. the maximizing $v^*$ indexes the largest $x_i$. Similarly, one may parameterize the choice of a permutation $P$ through a square matrix $X$, as the solution to the linear assignment problem (Kuhn, 1955), with $\mathcal{P}_N$ denoting the set of permutation matrices and $\langle A, B \rangle_F = \text{trace}(A^\top B)$ the (Frobenius) inner product of matrices:

$$
M(X) = \arg\max_{P \in \mathcal{P}_N} \langle P, X \rangle_F .
\tag{2}
$$

We call $M(\cdot)$ the matching operator, through which we parameterize the hard choice of a permutation (see Figure 3a for an example). Our theoretical contribution is to show that $M(X)$ can be obtained as the limit of $S(X/\tau)$, meaning that one can approximate $M(X) \approx S(X/\tau)$ with a small $\tau$. Theorem 1 summarizes our finding. We provide a rigorous proof in appendix A; briefly, it is based on showing that $S(X/\tau)$ solves a certain entropy-regularized problem in $\mathcal{B}_n$, which in the limit converges to the matching problem in equation 2.

**Theorem 1.** *For a doubly-stochastic matrix $P$, define its entropy as $h(P) = -\sum_{i,j} P_{i,j} \log(P_{i,j})$. Then, one has,*

$$
S(X/\tau) = \arg\max_{P \in \mathcal{B}_N} \langle P, X \rangle_F + \tau h(P).
\tag{3}
$$

---

[1]With the exception of the degenerate case of ties.

[2]This theorem requires certain technical conditions which are trivially satisfied if $X$ has positive entries, motivating the use of the component-wise exponential $\exp(\cdot)$ in the first line of equation 1.

*Now, assume also the entries of $X$ are drawn independently from a distribution that is absolutely continuous with respect to the Lebesgue measure in $\mathbb{R}$. Then, almost surely, the following convergence holds:*

$$M(X) = \lim_{\tau \to 0^+} S(X/\tau). \tag{4}$$

Finally, we note that Theorem 1 cannot be realized in practice, as it involves a limit on the Sinkhorn iterations $l$. Instead, we'll always consider the incomplete version of the Sinkhorn operator (Adams & Zemel, 2011), where we truncate $l$ in (1) to $L$. Figure 3b in appendix A.3 illustrates the dependence of the approximation in $\tau$ and $L$.

## 3 SINKHORN NETWORKS

Now we show how to apply the approximation in Theorem 1 in the context of artificial neural networks. We construct a layer that encodes the representation of a permutation, and show how to train networks containing such layers as intermediate representations.

We define the components of this network through a minimal example: consider the supervised task of learning a mapping from scrambled objects $\tilde{X}$ to actual, non-scrambled $X$. Data, then, are $M$ pairs $(X_i, \tilde{X}_i)$ where $\tilde{X}_i$ can be constructed by randomly permuting pieces of $X_i$. We state this problem as a permutation-valued regression $X_i = P_{\theta,\tilde{X}_i}^{-1} \tilde{X}_i + \varepsilon_i$, where $\varepsilon_i$ is a noise term, and $P_{\theta,\tilde{X}_i}$ is the permutation matrix mapping $X_i$ to $\tilde{X}_i$, which depends on $\tilde{X}_i$ and parameters $\theta$. We are concerned with minimization of the reconstruction error [3]:

$$f(\theta, X, \tilde{X}) = \sum_{i=1}^{M} ||X_i - P_{\theta,\tilde{X}_i}^{-1} \tilde{X}_i||^2. \tag{5}$$

One way to express a complex parameterization of this kind is through a neural network: this network receives $\tilde{X}_i$ as input, which is then passed through some intermediate, feed-forward computations of the type $g_h(W_h x_h + b_h)$, where $g_h$ are nonlinear activation functions, $x_h$ is the output of a previous layer, and $\theta = \{(W_h, b_h)\}_h$ are the network parameters. To make the final network output be a permutation, we appeal to constructions developed in Section 2: by assuming that the final network output $P_{\theta,\tilde{X}}$ can be parameterized as the solution of the assignments problem; i.e., $P_{\theta,\tilde{X}} = M(g(\tilde{X}, \theta))$, where $g(\cdot, \theta)$ represents the outcome of all operations involving $g_h$.

Unfortunately, the above construction involves a non-differentiable $f$ (in $\theta$). We use Theorem 1 as a justification for replacing $M(g(\tilde{X}, \theta))$ by the differentiable $S(g(\tilde{X}, \theta)/\tau)$ in the computational graph. The value of $\tau$ must be chosen with caution: if $\tau$ is too small, gradients vanishes almost everywhere, as $S(g(\tilde{X}, \theta)/\tau)$ approaches the non-differentiable $M(g(\tilde{X}, \theta))$. Conversely, if $\tau$ is too large, $S(X/\tau)$ may be far from the vertices of the Birkhoff polytope, and reconstructions $P_{\theta,\tilde{X}}^{-1} \tilde{X}$ may be nonsensical (see Figure 2a). Importantly, we will always add noise to the output layer $g(\tilde{X}, \theta)$ as a regularization device: by doing so we ensure uniqueness of $M(g(\tilde{X}, \theta))$, which is required for convergence in Theorem 1.

### 3.1 PERMUTATION EQUIVARIANCE

Among all possible architectures that respect the aforementioned parameterization, we will only consider networks that are *permutation equivariant*, the natural kind of symmetry arising in this context. Specifically, we require networks to satisfy:

$$P_{\theta,P'\tilde{X}}\left(P'\tilde{X}\right) = P'\left(P_{\theta,\tilde{X}}\tilde{X}\right)$$

where $P'$ is an arbitrary permutation. The underlying intuition is simple: reconstructions of objects should not depend on how pieces were scrambled, but only on the pieces themselves. We achieve permutation equivariance by using the same network to process each piece of $\tilde{X}$, throwing an $N$

---

[3]This error arises from gaussian $\varepsilon_i$. Other choices may be possible, but here we stick to the most straightforward formulation

dimensional output. Then, these $N$ outputs (each with $N$ components) are used to create the rows of the matrix $g(\tilde{X}, \theta)$, to which we finally apply the (differentiable) Sinkhorn operator (i.e. $g$ stacks the composition of the $g_h$ acting locally on each piece). One can interpret each row as representing a vector of local likelihoods of assignment, but they might be inconsistent. The Sinkhorn operator, then, mixes those separate representations, and ensures that consistent (approximate) assignment are produced. With permutation equivariance, the only consideration left to the practitioner is the choice of the particular architecture, which will depend on the particular kind of data. In Section 5 we illustrate the uses of Sinkhorn networks with three examples, each of them using a different architecture. Also, in figure 1 we illustrate a network architecture used in one of our examples.

## 3.2 SUMMARY

Sinkhorn network is a supervised method for learning to reconstruct a scrambled object $\tilde{X}$ (input) given several training examples $(X_i, \tilde{X}_i)$. By applying some non-linear transformations, a Sinkhorn network richly parameterizes the mapping between $\tilde{X}$ and the permutation $P$ that once applied to $\tilde{X}$, will allow to reconstruct the original object as $X_{rec} = P^\top \tilde{X}$ (the output). We note that Sinkhorn networks may be similarly used not only to learn permutations, but also to learn matchings between objects of two sets of the same size.

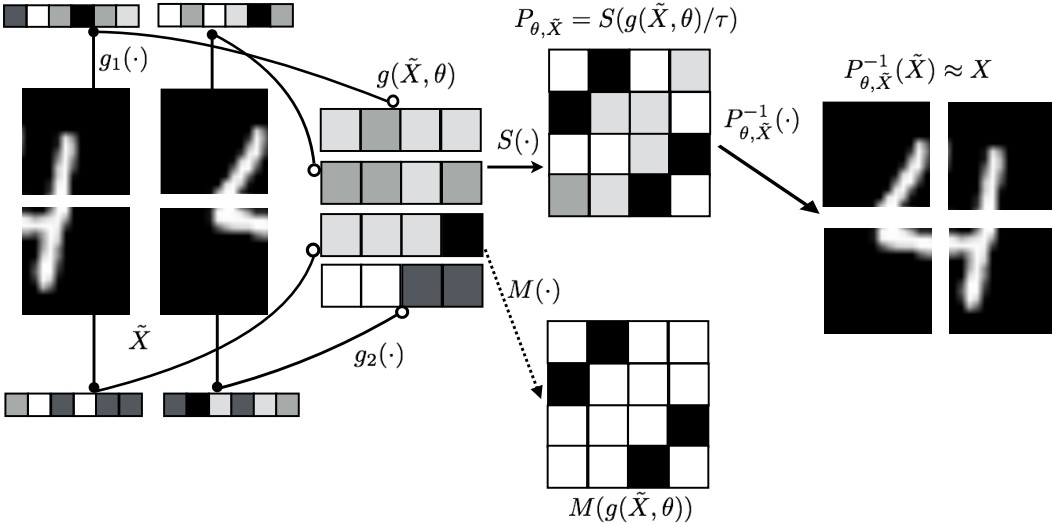

Figure 1: Schematic of Sinkhorn Network for Jigsaw puzzles. Each piece of the scrambled digit $\tilde{X}$ is processed with the same (convolutional) network $g_1$ (arrows with solid circles). The outputs lying on a latent space (rectangles surrounding $\tilde{X}$) are then connected through $g_2$ (arrows with empty circles) to conform the rows of the matrix $g(\tilde{X}, \theta)$; $g(\tilde{X}, \theta)_i = g_1 \circ g_2(\tilde{X}_i)$. Rows may be interpreted as unnormalized assignment probabilities, indicating individual unnormalized likelihoods of pieces of $\tilde{X}$ to be at every position in the actual image. Applying $S(\cdot)$ leads to a 'soft-permutation' $P_{\theta, \tilde{X}}$ that resolves inconsistencies in $g(\tilde{X}, \theta)$. $P_{\theta, \tilde{X}}$ is then used to recover the actual $X$ at training, although at test time one may use the actual $M(g(\tilde{X}, \theta))$.

## 4 PROBABILISTIC ASPECTS: THE GUMBEL-SINKHORN AND GUMBEL-MATCHING DISTRIBUTIONS

Recently, in Jang et al. (2016) and Maddison et al. (2016), the Gumbel-Softmax or Concrete distributions were defined for computational graphs with stochastic nodes; i.e, latent probabilistic representations. Their choice is guided by the following i) they seek re-parameterizable distributions to enable the re-parameterization trick (Kingma & Welling, 2013), and note that via the *Gumbel trick* (see below) any categorical distribution is re-parameterizable, ii) since the re-parameterization in i)

is not differentiable, they consider instead sampling under the softmax approximation. This gives rise to the Gumbel-Softmax distribution.

Here we parallel these choices to enable learning of a probabilistic latent representation of permutations. To this aim, we start by considering a generic distribution on the discrete set $\mathcal{Y}$, with potential function $X : \mathcal{Y} \rightarrow \mathbb{R}$:

$$p(y|X) \propto \exp\left(X(y)\right) \mathbf{1}_{y \in \mathcal{Y}}. \tag{6}$$

Regarding i), the Gumbel trick arises in the context of Perturb and MAP methods (Papandreou & Yuille, 2011) for sampling in discrete graphical models. This has recently received renewed interest (Balog et al., 2017), as it recasts the a difficult sampling problem as an easier optimization problem. In detail, sampling from (6), can be achieved by the maximization of random perturbations of each potential $X(y)$, with Gumbel i.i.d. noise $\gamma(y)$; i.e., $\arg\max_{y \in \mathcal{Y}}\{X(y) + \gamma(y)\} \sim p(\cdot|X)$. Therefore, one can re-parameterize any categorical distribution (corresponding to (6) with $X(y) = \langle X, y \rangle$) by the choice of a category, after injecting noise.

However, the above scheme is unfeasible in our context, as $|\mathcal{Y}| = N!$. Nonetheless, we appeal to an interesting result: in cases where $\mathcal{Y}$ factorizes, $\mathcal{Y} = \prod_{i=1}^{N} \mathcal{Y}_i$ [4], the use of rank-one perturbations $\gamma(y) = \sum_{i=1}^{N} \gamma_i(y_i)$ is proposed as a more tractable alternative. Although ultimately heuristic, they lead to bounds in the partition function (Hazan & Jaakkola, 2012; Balog et al., 2017), and can also be understood as providing approximate or unbiased samples from the true density (Hazan et al., 2013; Tomczak, 2016).

Guided by this, we say the random permutation $P$ follows the *Gumbel-Matching* distribution with parameter $X$, denoted $P \sim \mathcal{G}.\mathcal{M}.(X)$, if it has the distribution arising by the rank-one perturbation of (6) on permutations, with the linear potential $X(P) = \langle X, P \rangle_F$ (replacing $y$ with $P$). One can verify, in a similar line as in Li et al. (2013), that $M(X + \varepsilon) \sim \mathcal{G}.\mathcal{M}.(X)$, if $\varepsilon$ is a matrix of standard i.i.d. Gumbel noise.

Unfortunately, as ii) with the categorical case, Gumbel-Matching distribution samples are not differentiable in $X$, but by appealing to Theorem 1, we define its relaxation for doubly stochastic matrices as follows: we say $P$ follows the *Gumbel-Sinkhorn* distribution with parameter $X$ and temperature $\tau$, denoted $P \sim \mathcal{G}.\mathcal{S}.(X, \tau)$, if it has the distribution of $S((X + \varepsilon)/\tau)$. Samples of $\mathcal{G}.\mathcal{S}.(X, \tau)$ converge almost surely to samples of the Gumbel-Matching distribution (see Fig 3c in appendix A.3).

Unlike for the categorical case, neither the Gumbel-Matching nor Gumbel-Sinkhorn distributions have tractable densities. However, this does not preclude inference: likelihood-free methods have recently been developed to enable learning in such implicitly defined distributions (Ranganath et al., 2016; Tran et al., 2017). These methods avoid evaluating the likelihood based on the observation that in many cases inference can be cast as the estimation of a likelihood ratio, which can be obtained from samples (Huszár, 2017). Regardless of these useful advances, in the following we develop a solution based on using the likelihoods of random variables whose densities *are available*.

## 4.1 Approximate Posterior Inference

Consider a latent variable model probabilistic model with observed data $Y$, and latent $Z = \{P, W\}$ where $P$ is a permutation and $W$ are other variables. Here we illustrate how to approximate the posterior probability $p(\{P, W\}|Y)$ using variational inference Blei et al. (2017). Specifically, we aim to maximize the ELBO, the r.h.s. of (7):

$$\log p(y) \geq E_{q(Z|Y)}\left(\log p(Y|Z)\right) - KL(q(Z|Y) \parallel p(Z)). \tag{7}$$

We assume that both the prior and variational posteriors decompose as products (mean-field). That is, $q(\{P, W\}|Y) = q(P)q(W), p(P, W) = p(P)p(W)$. With this assumption, we may focus only on the discrete part of the problem, i.e. without loss of generality we can assume $Z = P$.

We parameterize our variational prior and posteriors on $P$ using the Gumbel-Matching distributions with some parameter $X$; $\mathcal{G}.\mathcal{M}.(X)$. To enable differentiability, we replace them by $\mathcal{G}.\mathcal{S}.(X, \tau)$ distributions, leading to a surrogate ELBO that uses relaxed (continuous) variables. In more detail,

---

[4]It suffices that $\mathcal{Y}$ is a subset of the product space, which here is true as $\mathcal{Y} = \mathcal{P}_n \subseteq \{1, \ldots, N\}^N$.

| Test distribution | $N = 5$ | $N = 10$ | $N = 15$ | $N = 80$ | $N = 100$ | $N = 120$ |
|---|---|---|---|---|---|---|
| $U(0, 1)$ | **.0** | **.0** | **.0** | **.0** | **.0** | **.01** |
| $U(0, 1)$ (Vinyals et al., 2015) | .06 | 0.43 | 0.9 | - | - | - |
| $U(0, 10)$ | .0 | .0 | .0 | .0 | .02 | .03 |
| $U(0, 1000)$ | .0 | .0 | .0 | .01 | .02 | .04 |
| $U(1, 2)$ | .0 | .0 | .0 | .01 | .04 | .08 |
| $U(10, 11)$ | .0 | .0 | .0 | .08 | .08 | .6 |
| $U(100, 101)$ | .0 | .0 | .01 | .02 | .99 | 1. |
| $U(1000, 1001)$ | .0 | .0 | .07 | 1. | 1. | 1. |

Table 1: Results on the number sorting task measured using Prop. any wrong. In the top two rows we compare to Vinyals et al. (2015), showing that our approach can sort far more inputs at significantly higher accuracy. In the bottom rows we evaluate generalization to different intervals on the real line.

for our uniform prior over permutations we use the isotropic $\mathcal{G}.\mathcal{S}.(X = 0, \tau_{prior})$ distribution, while for the variational posterior we consider the more generic $\mathcal{G}.\mathcal{S}.(X, \tau)$.

Unfortunately, the term $KL(q(P|Y) \parallel p(P)) = KL(\mathcal{G}.\mathcal{S}.(X, \tau) \parallel \mathcal{G}.\mathcal{S}.(X = 0, \tau_{prior}))$ in equation (7) is intractable as there is not closed form expression for the density of $\mathcal{G}.\mathcal{S}.$ random variables. As a solution, we use that our prior and posterior are re-parameterizable in terms of matrices $\varepsilon$ of Gumbel i.i.d variables: we have $S((X + \varepsilon)/\tau) \sim \mathcal{G}.\mathcal{S}.(X, \tau)$ and $S(\varepsilon/\tau_{prior}) \sim \mathcal{G}.\mathcal{S}.(X = 0, \tau_{prior})$, for the posterior and prior, respectively. To obtain a tractable expression, we propose to use as 'code' or stochastic node $Z$, the variable $(X + \varepsilon)/\tau$ instead. Then, the KL term substantially simplifies to $KL((X + \varepsilon)/\tau \parallel \varepsilon/\tau_{prior})$. This term can be computed explicitly, as shown in appendix B.3.

This 'trick', however, comes at a cost: the divergence $KL(Z_1 \parallel Z_2)$ would certainly remain unchanged by applying the same invertible transformation $g$ to both variables $Z_1$ and $Z_2$, but in the general case, for non-invertible transformations, such as $S(\cdot)$, one has $KL(Z_1 \parallel Z_2) \geq KL(g(Z_1) \parallel g(Z_2))$. This implies that working in the 'Gumbel space' might entail the optimization of a less tight lower bound. Nonetheless, through categorical experiments on MNIST (see appendix C.3) we observe this loss of tightness is minimal, suggesting the suitability of our approach on permutations. Finally, we note that key to to our treatment of the problem is the fact that both the prior and posterior were the same function $(S(\cdot))$ of a simpler distribution. This may not be the case in more general models.

To conclude this section, we refer the reader to table 8 in appendix D.2 for a summary of all the constructions on permutations developed in this work.

## 5 EXPERIMENTS

In this section we perform several experiments comparing to existing methods. In the first three experiments we explore different Sinkhorn network architectures of increasing complexity, and therefore, they mostly implements section 3. The fourth experiment relates to the probabilistic constructions described in section 4, and addresses a problem involving marginal inferences over a latent, unobserved permutation. All experimental details not stated here are in appendix B.

### 5.1 SORTING NUMBERS

To illustrate the capabilities of Sinkhorn Networks in a simple scenario, we consider the task of sorting numbers using artificial neural networks as in Vinyals et al. (2015). Specifically, we sample uniform random numbers $\tilde{X}$ in the $[0, 1]$ interval and we train our network with pairs $(\tilde{X}, X)$ where $X$ are the same $\tilde{X}$ but in sorted order. The network has a first fully connected layer that links a number with an intermediate representation (with 32 units), and a second (also fully connected) layer that turns that representation into a row of the matrix $g(\tilde{X}, \theta)$.

Table 1 shows our network learns to sort up to $N = 120$ numbers. As an evaluation measure, we report the proportion of sequences where there was at least one error (Prop. any wrong). Surprisingly,

| | MNIST | | | | | Celeba | | | | Imagenet | |
|---|---|---|---|---|---|---|---|---|---|---|---|
| | 2x2 | 3x3 | 4x4 | 5x5 | 6x6 | 2x2 | 3x3 | 4x4 | 5x5 | 2x2 | 3x3 |
| Kendall tau | 1. | .83 | .43 | .39 | .27 | 1.0 | .96 | .88 | .78 | .85 | **.73** |
| Kendall tau (Cruz et al., 2017) | - | - | - | - | - | - | - | - | - | - | .72 |
| Prop. wrong | .0 | .09 | .45 | .45 | .59 | .0 | .03 | .1 | .21 | .12 | .26 |
| Prop. any wrong | .0 | .28 | .97 | 1. | 1. | .0 | .09 | .36 | .73 | .19 | .53 |
| $l1$ | .0 | .0 | .04 | .02 | .03 | .0 | .01 | .04 | .08 | .05 | .12 |
| $l2$ | .0 | .0 | .26 | .18 | .19 | .0 | .11 | .18 | .24 | .22 | .31 |

Table 2: Jigsaw puzzle results. We compare to the available result on the Kendall Tau metric from Cruz et al. (2017) and provide additional results from our experiments. Randomly guessed permutations of $n$ items have an expected proportion of errors of $(n-1)/n$. Note that our model has at least 20x fewer parameters..

the network learns to sort numbers even when test examples are not sampled from $U(0,1)$, but on a considerably different interval. This indicates the network is not overfitting. These results can be compared with those from Vinyals et al. (2015), where a much more complex (recurrent) network was used, but performance guarantees were obtained only with at most $N = 15$ numbers. In that case, the reported error rate is 0.9, whereas ours starts to degrade only after $N \approx 100$ for most test intervals.

## 5.2 JIGSAW PUZZLES

A more complex scenario for learning permutations arises in the reconstruction of an image $X$ from a collection of scrambled "jigsaw" pieces $\tilde{X}$ (Noroozi & Favaro, 2016; Cruz et al., 2017). In this example, our network differs from the one in 5.1 in the first layer is a simple CNN (convolution + max pooling), which maps the puzzle pieces to an intermediate representation (see figure 1 for details).

For evaluation on test data, we report several measures: first, in addition to Prop. any wrong we also consider Prop. wrong, the overall proportion of scrambled pieces that were wrongly assigned to their actual position. Also, we use $l1$ and $l2$ (train) losses and the Kendall tau, a "correlation coefficient" for ranked data. In Table 2, we benchmark results for the MNIST, Celeba and Imagenet datasets, with puzzles between 2x2 and 6x6 pieces. In MNIST we achieve very low $l1$ and $l2$ on up to 6x6 puzzles but a high proportion of errors. This is a consequence of our loss being agnostic to particular permutations, but only caring about reconstruction errors: as the number of black pieces increases with the number of puzzle pieces, many become unidentifiable under this loss.

In Celeba, we are able to solve puzzles of up to 5x5 pieces with only 21% of pieces of faces being incorrectly ordered (see Figure 2a for examples of reconstructions). For this dataset, we provide additional baselines in Table 4 of appendix C.1: there, we show that performance substantially decreases if the temperature is too small or large, but only slightly decreases if only one Sinkhorn iterations is made. We observe that temperature does play a relevant role, consistent with the findings of Maddison et al. (2016); Jang et al. (2016). This might not be obvious a-priori, as one could reason that temperature over-parameterizes the network. However, results confirm this is not the case. We hypothesize that different temperatures result in parameter convergence in different phases or regions. Also, the minor difference for a single iteration suggest that only a few might be necessary, implying potential savings in the memory needed to unroll computations in the graph, during training.

Learning in the Imagenet dataset is much more challenging, as there isn't a sequential structure that generalizes among images, unlike Celeba and MNIST. In this dataset, our network ties with the .72 Kendall tau score reported in (Cruz et al., 2017). Their network, named DeepPermNet, is based on the stacking of up to the sixth fully connected layer *fc6* of AlexNet (Krizhevsky et al., 2012), which finally (fully) connects to a Sinkhorn layer through intermediate *fc7* and *fc8*. We note, however, our network is much simpler, with only two layers and far fewer parameters. Specifically, the network

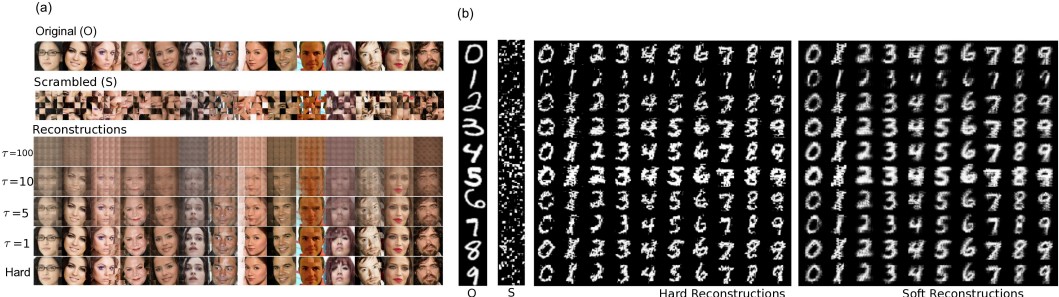

Figure 2: (a) Sinkhorn networks can be trained to solve Jigsaw Puzzles. Given a trained model, 'soft' reconstructions are shown at different $\tau$ using $S(X/\tau)$. We also show hard reconstructions, made by computing $M(X)$ with the Hungarian algorithm (Munkres, 1957). (b) Sinkhorn networks can also be used to learn to transform any MNIST digit into another. We show hard and soft reconstructions, with $\tau = 1$.

that produced our best results had around 1,050,000 parameters (see appendix B for a derivation), while in DeepPermNet, the layer connecting *fc6* with *fc7* has $512 \times 4096 \times 9 \approx 19,000,000$ parameters, let alone the AlexNet parameters (also to be learned). Indeed, we believe there is no reason to consider a complex stacking of convolutions: as the number of pieces increases, each piece is smaller and the convolutional layer eventually becomes fully connected. In the following experiment we explore this phenomenon in more detail.

## 5.3 ASSEMBLY OF ARBITRARY MNIST DIGITS FROM PIECES

We also consider an original application, motivated by the observation that the Jigsaw Puzzle task becomes ill-posed if a puzzle contains too many pieces. Indeed, consider the binarized MNIST dataset: there, reconstructions are not unique if pieces are sufficiently atomic, and in the limit case of pieces of size 1x1 squared pixels, for a given scrambled MNIST digit there are as many valid reconstructions as there are MNIST digits with the same number of white pixels. In other words, reconstructions stop being probabilistic and become a multimodal distribution over permutations.

We exploit this intuition to ask whether a neural network can be trained to achieve arbitrary digit reconstructions, given their loose atomic pieces. To address this question, we slightly changed the network in 5.2, this time stacking several second layers linking an intermediate representation to the output. We trained the network to reconstruct a particular digit with each layer, by using digit identity to indicate which layer should activate with a particular training example.

Our results demonstrate a positive answer: Figure 2b shows reconstructions of arbitrary digits given 10x10 scrambled pieces. In general, they can be unambiguously identified by the naked eye. Moreover, this judgement is supported by the assessment of a neural network. Specifically, we trained a two-layer CNN [5] on MNIST (achieving a 99.2% accuracy on test set) and evaluated its performance on the test set generated by arbitrary transformations of each digit of the original test set into any other digit. We found the CNN made an appropriate judgement in 85.1% of the time. More specific results, regarding specific transformations are presented in Table 5 of appendix C.2.

Finally, we note that meaningful assemblies are possible regardless of the original digit: in Figure 4 of appendix C.2 we show arbitrary reconstructions, by this same network, of "digits" from a 'strongly mixed' MNIST dataset. In detail, these "digits" were crafted by sampling, without replacement, from a bag containing all the small pieces from all original digits. These reconstructions suggest the possibility of an alternative to generative modeling, based on the (random) assembly of small pieces of noise, instead of the processing of noise through a neural network. However, this would require training the network without supervision, which is beyond the scope of this work.

---

[5]Specifically, we used the one described in the Deep MNIST for experts tutorial.

| Prop. known neurons | 40.% | | 30.% | | 20.% | | 10.% | |
|---|---|---|---|---|---|---|---|---|
| Difficulty | Easy | Hard | Easy | Hard | Easy | Hard | Easy | Hard |
| MCMC | .85 | .82 | .51 | .44 | .29 | .27 | .16 | .12 |
| (Linderman et al., 2017) | **.97** | .95 | .90 | **.85** | **.77** | **.59** | .39 | .21 |
| Gumbel-Sinkhorn | **.97** | **.96** | **.92** | .84 | .76 | **.59** | **.44** | **.26** |
| Gumbel-Sinkhorn, no regularization | .96 | .93 | .89 | .78 | .71 | .52 | .4 | .23 |

Table 3: Results for the C. elegans neural inference problem.

## 5.4 POSTERIOR INFERENCE OVER PERMUTATIONS WITH THE GUMBEL-SINKHORN ESTIMATOR

We illustrate how the $\mathcal{G.S.}$ distribution can be used as a continuous relaxation for stochastic nodes in a computational graph. To this end, we revisit the "C. elegans neural identification problem", originally introduced in Linderman et al. (2017). We refer the reader to (Linderman et al., 2017) for an in-depth introduction, but briefly, *C. elegans* is a nematode (worm) whose biological neural configuration – the *connectome* – is stereotypical; i.e. specimens always posses the same number of somatic neurons (282) (Varshney et al., 2011), and the ways those neurons connect and interact changes little from worm to worm. Therefore, its brain can be thought of as a canonical object, and its neurons can unequivocally be identified with names.

The task, then, consists of matching traces from the observed neural dynamics $Y$ to identities (neuron names) in the canonical brain. This problem is stated in terms of a Bayesian hierarchical model, in order to profit from prior information that may constrain the possibilities. Specifically, one states a linear dynamical system $Y_t = PWP^\mathsf{T}Y_{t-1} + \nu_t$, where $\nu_t$ is a noise term and $W$ and $P$ are latent variables with respective prior distributions. $W$ encodes the dynamics, with a prior $p(W)$ to represent the sparseness of the connectome, etc., and $P$ is a permutation matrix representing the matching between indexes of observed neurons and their canonical counterparts, where we place a flat prior $p(P)$ over permutations. Notably, within the framework it is possible to model the simultaneous problem with many worms sharing the same dynamical system, but here we avoid explicit references to individuals for notational ease.

Given this model, we seek the posterior distribution $p(\{P, W\}|Y)$, a problem that we address with variational inference (Blei et al., 2017) using the constructions developed in 4.1. In Table 3 (and also in Table 7 of appendix C.4) we show results for this task, using accuracy in matching as the performance measure. These are broken down by relevant experimental covariates (Linderman et al., 2017): different proportion of neurons known beforehand, and by task difficulty. As baselines, we include i) a simple MCMC sampler that proposes local swipes on permutations ii) the rounding method presented in Linderman et al. (2017), iii) our method, where we also consider the absence of regularization. Results show our method outperforms the alternatives in most cases. MCMC fails because mixing is poor, but differences are much subtler with the other baselines. With them, we see that clear differences with the no-regularization case confirm the stochastic nature of this problem, i.e., that it is truly necessary to represent a latent probabilistic permutation. We believe our method outperforms the one in Linderman et al. (2017) because theirs, although it provides a explicit density, is a less tight relaxation, in the sense that points can be anywhere in the space, and not only on the Birkhoff polytope. Therefore, their prior also needs to be defined on the entire space and may not property act as an efficient regularizer.

## 6 RELATED WORK

Learning with matchings has been extensively been studied in the machine learning community; but current applications mostly relate to structured prediction (Petterson et al., 2009; Tang et al., 2016). However, our probabilistic treatment focuses on marginal inference in a model with a latent matching. This is a more challenging scenario, as standard learning techniques, i.e. the score function estimator or REINFORCE (Williams, 1992), are not applicable due to the partition function for non-trivial distributions over matchings.

In the case of latent categories, a recent technique that combines a relaxation and the re-parameterization trick (Kingma & Welling, 2013) was proposed as a competitive alternative to RE-INFORCE for the marginal inference scenario. Specifically, Maddison et al. (2016); Jang et al. (2016) use the Gumbel-trick to re-parameterize a discrete density, and then replace it with a re-laxed surrogate, the Gumbel Softmax distribution, to enable gradient-descent. Our work, like the simultaneous work of Linderman et al. (2017), aims to extends the scope of this technique to latent permutations. We deem our Gumbel Sinkhorn distributions as the most natural tractable extension of the Gumbel Softmax to permutations, as we clearly parallel each of the steps leading to its con-struction. A parallel is also presented in Linderman et al. (2017); and notably, unlike ours, their framework produces tractable densities. However, it is less clear how their constructions extend each of the features of the Gumbel Softmax: for example, their rounding-based relaxation also utilizes the Sinkhorn operator, but the limit they consider does not make use of the non-trivial statement of Theorem 1, which naturally extends the categorical case (see appendix A.2 for details). In practice, we see our results favor the Gumbel Sinkhorn distribution, since it is a tighter relaxation.

Connections between permutations and the Sinkhorn operator have been known for at least twenty years. Indeed, the limit in Theorem 1 was first presented in Kosowsky & Yuille (1994), but their interpretation and motivation were more linked to statistical physics and economics. However, our approach is different and links to recent developments in optimal transport (OT) (Villani, 2003): Theorem 1 draws on the entropy-regularization for OT technique developed inCuturi (2013), where the entropy-regularized transportation problem is referred to as a 'Sinkhorn distance'. The exten-sion is sensible as in the case of transportation between two discrete measures (here) the Birkhoff polytope appears naturally as the optimization set (Villani, 2003). Entropy regularization as means to achieve a differentiable version of a loss was first proposed in Genevay et al. (2017) in the con-text of generative modeling. Although this field may appear separate, recent work (Salimans et al., 2018) makes explicit the connection to permutations: to compute a (Wasserstein) distance between a batch of dataset samples and one of generative samples of the same size, one needs to solve the matching problem so that the distance between matched samples is minimized. Finally, we note our work shares with Salimans et al. (2018); Genevay et al. (2017) in that the OT cost function (here, the matrix $X$) is learned using an artificial neural network.

We understand our work as extending Adams & Zemel (2011), which developed neural networks to learn a permutation-like structure; a ranking. However, there, as in Helmbold & Warmuth (2009), the objective function was linear and the Sinkhorn operator was instead used as an approximation of a matrix of the marginals, i.e., $S(P) \approx E(P)$. In consequence, there was no need to introduce a temperature parameter and consider a limit argument, which is critical to our case. Interestingly, equation (10) can be understood in terms of approximate marginal inference, justifying the approx-imation $S(P) \approx E(P)$. We comment on this in appendix D.1. Note that Sinkhorn iteration can be interpreted as mean-field inference in an associated Gibbs distribution over matchings. With this in mind, backpropagation through Sinkhorn is an end-to-end learning in an unrolled inference algorithm Stoyanov et al. (2011); Domke (2013). In future work, it may be fruitful to unroll alterna-tive algorithms for marginal inference over matchings, such as belief propagation (Huang & Jebara, 2009).

Sinkhorn networks were also very recently introduced in Cruz et al. (2017), although their work substantially differs from ours. While their interest lies in the representational aspects of CNN's, we are more concerned with the more fundamental properties. In their work, they don't consider a temperature parameter $\tau$, but their network still successfully learns, as $\tau = 1$ happens to fall within the range of reasonable values. On the Jigsaw puzzle task, we showed that we achieve equivalent performance with a much simpler network having several times fewer parameters and layers. Nonetheless, we recognize the need for more complex architectures for the tasks considered in Cruz et al. (2017), and we hope our more general theory; particularly, Theorem 1 and the notion of equivariance, may aid further developments in that direction.

## 7 DISCUSSION

We have demonstrated Sinkhorn networks are able to learn to find the right permutation in the most elementary cases; where all training samples obey the same sequential structure; e.g., in sorted number and in pieces of faces, as we expect parts of faces occupy similar positions from sample to

sample. This is already non-trivial, as indicates one can train a neural network to solve the linear assignment problem.

However, the fact that Imagenet represented a much more challenging scenario indicates there are clear limits to our formulation. As the most obvious extension we propose to introduce a sequential stage, in which current solutions are kept on a memory buffer, and improved. One way to achieve this would be by exploring more complex parameterizations for permutations; i.e. replacing $M(X)$ by a quadratic operator that may parameterize a notion of local distance between pieces. Alternatively, one may resort to reinforcement learning techniques, as suggested in Bello et al. (2016). Either sequential improvement would help solve the "Order Matters" problem (Vinyals et al., 2015), and we deem our elementary work as a significant step in that direction.

We have made available Tensorflow code for Gumbel-Sinkhorn networks featuring an implementation of the number sorting experiment at http://github.com/google/gumbel_sinkhorn .

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

# A    PROOF OF THEOREM 1

In this section we give a rigorous proof of Theorem 1. Also, in A.2 we briefly comment on how Theorem 1 extend a perhaps more intuitive results, in the probability simplex.

Before stating Theorem 1 we need some preliminary definitions. We start by recalling a well-known result in matrix theory, the Sinkhorn theorem.

**Theorem (Sinkhorn).** *Let $A$ be an $N$ dimensional square matrix with positive entries. Then, there exists two diagonal matrices $D_1, D_2$, with positive diagonals, so that $P = D_1 A D_2$ is a doubly stochastic matrix. These $D_1, D_2$ are unique up to a scalar factor. Also, $P$ can be obtained through the iterative process of alternatively normalizing the rows and columns of $A$.*

*Proof.* See Sinkhorn (1964); Sinkhorn & Knopp (1967); Knight (2008). ◻

For our purposes, it is useful to define the Sinkhorn operator $S(\cdot)$ as follows:

**Definition 1.** *Let $X$ be an arbitrary matrix with dimension $N$. Denote $\mathcal{T}_r(X) = X \oslash (X 1_N 1_N^\top)$, $\mathcal{T}_c(X) = X \oslash (1_N 1_N^\top X)$ (with $\oslash$ representing the element-wise division and $1_n$ the $n$ dimensional vector of ones) the row and column-wise normalization operators, respectively. Then, we define the Sinkhorn operator applied to $X$; $S(X)$, as follows:*

$$
\begin{aligned}
S^0(X) &= \exp(X), \\
S^l(X) &= \mathcal{T}_c\left(\mathcal{T}_r(S^{l-1}(X))\right), \\
S(X) &= \lim_{n \to \infty} S^l(X).
\end{aligned}
$$

*Here, the $\exp(\cdot)$ operator is interpreted as the component-wise exponential. By Sinkhorn's theorem, $S(X)$ is a doubly stochastic matrix.*

Finally, we review some key properties related to the space of doubly stochastic matrices. First, we need to define a relevant geometric object.

**Definition 2.** *We denote by $\mathcal{B}_N$ the $N$-Birkhoff polytope, i.e., the set of doubly stochastic matrices of dimension $N$. Likewise, we denote $\mathcal{P}_n$ be the set of permutation matrices of size $N$. Alternatively,*

$$
\mathcal{B}_N = \{P \in [0,1] \in \mathbb{R}^{N,N} \; P 1_N = 1_N, P^\top 1_N = 1_N\},
$$

$$
\mathcal{P}_N = \{P \in \{0,1\} \in \mathbb{R}^{N,N} \; P 1_N = 1_N, P^\top 1_N = 1_N\}.
$$

**Theorem (Birkhoff).** *$\mathcal{P}_N$ is the set of extremal points of $\mathcal{B}_N$. In other words, the convex hull of $\mathcal{B}_N$ equals $\mathcal{P}_N$.*

*Proof.* See Birkhoff (1946). ◻

## A.1    AN APPROXIMATION THEOREM FOR THE MATCHING PROBLEM

Let's now focus on the standard combinatorial assignment (or matching) problem, for an arbitrary $N$ dimensional matrix $X$. We aim to maximize a linear functional (in the sense of the Frobenius norm) in the space of permutation matrices. In this context, let's define the matching operator $M(\cdot)$ as the one that returns the solution of the assignment problem:

$$
M(X) \equiv \arg\max_{P \in \mathcal{P}_N} \langle P, X \rangle_F. \tag{8}
$$

Likewise, we define $\tilde{M}(\cdot)$ as a related operator, but changing the feasible space by the Birkhoff polytope:

$$
\tilde{M}(X) \equiv \arg\max_{P \in \mathcal{B}_N} \langle P, X \rangle_F. \tag{9}
$$

Notice that in general $\tilde{M}(X), M(X)$ might not be unique matrices, but a face of the Birkhoff polytope, or a set of permutations, respectively (see Lemma 2 for details). In any case, the relation

$M(X) \subseteq \tilde{M}(X)$ holds by virtue of Birkhoff's theorem, and the fundamental theorem of linear programming.

Now we state the main theorem of this work:

**Theorem 1.** *For a doubly stochastic matrix $P$ define its entropy as $h(P) = -\sum_{i,j} P_{i,j} \log(P_{i,j})$. Then, one has,*

$$S(X/\tau) = \arg\max_{P \in \mathcal{B}_N} \langle P, X \rangle_F + \tau h(P). \tag{10}$$

*Now, assume also the entries of $X$ are drawn independently from a distribution that is absolutely continuous with respect to the Lebesgue measure in $\mathcal{R}$. Then, almost surely the following convergence holds:*

$$M(X) = \lim_{\tau \to 0^+} S(X/\tau). \tag{11}$$

We divide the proof of Theorem 1 in three steps. First, in Lemma 1 we state a relation between $S(X/\tau)$ and the entropy regularized problem in equation (10). Then, in Lemma 2 we show that under our stochastic regime, uniqueness of solutions holds. Finally, in Lemma 3 we show that in this well-behaved regime, convergence of solutions holds. states that and Lemma 2b endows us with the tools to make a limit argument.

### A.1.1 INTERMEDIATE RESULTS FOR THEOREM 1

**Lemma 1.**

$$S(X/\tau) = \arg\max_{P \in \mathcal{B}_N} \langle P, X \rangle_F + \tau h(P).$$

*Proof.* We first notice that the solution $P_\tau$ of the above problem exists, and it is unique. This is a simple consequence of the strict concavity of the objective (recall the entropy is strictly concave Rao (1984)).

Now, let's state the Lagrangian of this constrained problem

$$\mathcal{L}(\alpha, \beta, P) = \langle P, X \rangle_F + \tau h(P) + \alpha^\top (P 1_N - 1_N) + \beta^\top (P^\top 1_N - 1_N),$$

It is easy to see, by stating the equality $\partial \mathcal{L}/\partial P = 0$ that one must have for each $i, j$,

$$p_\tau^{i,j} = \exp(\alpha_i/\tau - 1/2) \exp(X_{i,j}/\tau) \exp(\beta_j/\tau - 1/2),$$

in other words, $P_\tau = D_1 \exp(X_{i,j}/\tau) D_2$ for certain diagonal matrices $D_1, D_2$, with positive diagonals. By Sinkhorn's theorem, and our definition of the Sinkhorn operator, we must have that $S(X/\tau) = P_\tau$. $\square$

**Lemma 2.** *Suppose the entries of $X$ are drawn independently from a distribution that is absolutely continuous with respect to the Lebesgue measure in $\mathbb{R}$. Then, almost surely, $\tilde{M}(X) = M(X)$ is a unique permutation matrix.*

*Proof.* This is a known result from sensibility analysis on linear programming which we prove for completeness. Notice first that the problem in (2) is a linear program on a polytope. As such, by the fundamental theorem of linear program, the optimal solution set must correspond to a face of the polytope. Let $\mathcal{F}$ be a face of $\mathcal{B}_N$ of dimension $\geq 1$, and take $P_1, P_2 \in \mathcal{F}$, $P_1 \neq P_2$. If $\mathcal{F}$ is an optimal face for a certain $X_{\mathcal{F}}$, then $X_{\mathcal{F}} \in \{X : \langle P_1, X \rangle_F = \langle P_2, X \rangle_F\}$. Nonetheless, the latter set does *not* have full dimension, and consequently has measure zero, given our distributional assumption on $X$. Repeating the argument for every face of dimension $\geq 1$ and taking a union bound we conclude that, almost surely, the optimal solution lies on a face of dimension 0, i.e, a vertex. From here uniqueness follows. $\square$

**Lemma 3.** *Call $P_\tau$ the solution to the problem in equation 10, i.e. $P_\tau = P_\tau(X) = S(X/\tau)$. Under the assumptions of Lemma 2, $P_\tau \to P_0$ when if $\tau \to 0^+$.*

*Proof.* **Proof** Notice that by Lemmas 1 and 2, $P_\tau$ is well defined and unique for each $\tau \geq 0$. Moreover, at $\tau = 0$, $P_0 = M(X)$ is the unique solution of a linear program. Now, let's define $f_\tau(\cdot) = \langle \cdot, X \rangle_F + \tau h(\cdot)$. We observe that $f_0(P_\tau) \to f_0(P_0)$. Indeed, one has:

$$
\begin{aligned}
f_0(P_0) - f_0(P_\tau) &= \langle P_0, X \rangle_F - \langle P_\tau, X \rangle_F \\
&= \langle P_0, X \rangle_F - f_\tau(P_\tau) + \tau h(P_\tau) \\
&< \langle P_0, X \rangle_F - f_\tau(P_0) + \tau h(P_\tau) \\
&< \tau \left( h(P_\tau) - h(P_0) \right) \\
&< \tau \max_{P \in \mathcal{B}_N} h(P).
\end{aligned}
$$

From which convergence follows trivially. Moreover, in this case convergence of the values implies the converge of $P_\tau$: suppose $P_\tau$ does not converge to $P_0$. Then, there would exist a certain $\delta$ and sequence $\tau_n \to 0$ such that $\|P_{\tau_n} - P_0\| > \delta$. On the other hand, since $P_0$ is the unique maximizer of an LP, there exists $\varepsilon > 0$ such that $f_0(P_0) - f_0(P) > \varepsilon$ whenever $\|P - P_0\| > \delta$, $P \in \mathcal{B}_N$. This contradicts the convergence of $f_0(P_{\tau_n})$. $\square$

### A.1.2 PROOF OF THEOREM 1

The first statement is Lemma 1. Convergence (equation 11) is a direct consequence of Lemma 3, after noticing $P_\tau = S(X/\tau)$ and $P_0 = M(X)$. We note that an alternative approach for the limiting argument is presented in Cominetti & San Martín (1994).

### A.2 RELATION TO SOFTMAX

Finally, we notice that all of the above results can be understood as a generalization of the well-known approximation result $\arg\max_i x_i = \lim_{\tau \to 0^+} softmax(x/\tau)$. To see this, treat a category as a one-hot vector. Then, one has

$$
\arg\max_i x_i = \arg\max_{e \in \mathcal{S}_N} \langle e, x \rangle, \tag{12}
$$

where $\mathcal{S}_n$ is the probability simplex, the convex hull of the one-hot vectors (denoted $\mathcal{H}_n$). Again, by the fundamental theorem of linear algebra, the following holds:

$$
\arg\max_i x_i = \arg\max_{e \in \mathcal{H}_N} \langle e, x \rangle. \tag{13}
$$

On the other hand, by a similar (but simpler) argument than of the proof of theorem 4 one can easily show that

$$
softmax(x/\tau) \equiv \frac{\exp(x/\tau)}{\sum_{i=1} \exp(x_i/\tau)} = \arg\max_{e \in \mathcal{S}_n} \langle e, x \rangle + \tau h(e), \tag{14}
$$

where the entropy $h(\cdot)$ is not defined as $h(e) = -\sum_{i=1}^n e_i \log(e_i)$

### A.3 ILLUSTRATING THEOREM 1

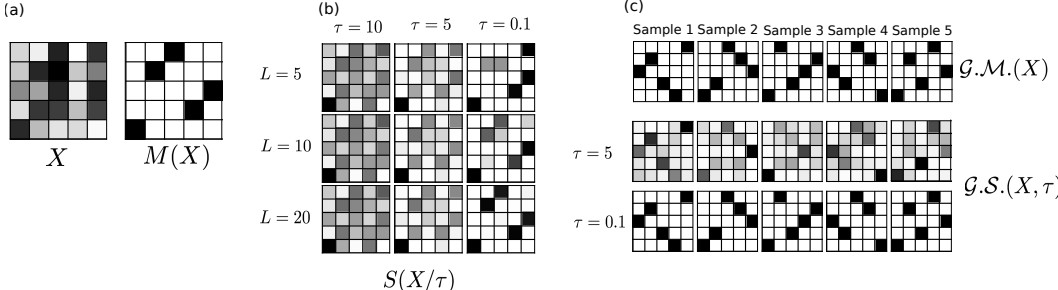

Figure 3: Illustrating the Matching and Sinkhorn operators, and the Gumbel-Matching and Gumbel-Sinkhorn distributions. Each 5x5 grid represents a matrix, with the shading indicating cell values (a) Matching operator $M(X)$ applied to a parameter matrix $X$. (b) Sinkhorn Operator $S(X/\tau)$ approximating $M(X)$ for different temperature $\tau$ and number of Sinkhorn iterations, $L$. (c). First row: samples from the Matching Sinkhorn distribution. Second and third rows: samples from the Gumbel-Sinkhorn distribution at two temperatures. At low temperature, both distributions are indistinguishable.

# B  SUPPLEMENTAL METHODS

## B.1  EXPERIMENTAL PROTOCOLS

All experiments were run on a cluster using Tensorflow Abadi et al. (2016), using several GPU (Tesla K20, K40, K80 and P100) in parallel to enable an efficient exploration of the hyperparameter space: temperature, learning rate, and neural network parameters (dimensions).

In all cases, we used $L = 20$ Sinkhorn Operator Iterations, and a 10x10 batch size: for each sample in the batch we used Gumbel perturbations to generate 10 different reconstructions.

For evaluation, we used the Hungarian Algorithm Munkres (1957) to compute $M(X)$ required to infer the predicted matching.

Finally, experiments of section 5.4 were done consistent with model specifications stated in Linderman et al. (2017)

## B.2  NUMBER OF PARAMETERS ON SINKHORN NETWORKS

In the simplest network, the one that sorts number, the number of parameters is given by $n_u + N \times n_u$: Indeed, each number is connected with the hidden layer with $n_u$ (here, 32) units. This layer connects with another layer with $N$ units, representing a row of $g(\tilde{X}, \theta)$.

For images, the first layer is a convolution, composed by $n_f$ convolutional filters of receptive field size $K_s$ with $n_c$ channels (one or three) followed by a ReLU + max-pooling (with stride $s$) operations. Then, the number of parameters in the first layer is given by $K_s^2 \times n_c \times n_f + n_f$. The second layers connects the output of a convolution, i.e., the stacked convolved $l \times l$ images by each of the filters (after max-pooling) and $p^2$ units, where $p$ is the number of pieces each side was divided by. Therefore, the number of parameters is given by $l^2/(p^2 s^2) \times n_f \times p^2 = l^2/s^2 \times n_f$, up to rounding and padding subtleties. Then, the total number of parameters is $l^2/s^2 \times n_f + K_s^2 \times n_c \times n_f + n_f$. For the 3x3 puzzle on Imagenet, $l = 256, p = 3, n_c = 3$ and the optimal network was such that $n_f = 64, s = 2, K_s = 5$. Then, it had 1,053,440 parameters.

Finally, for arbitrary assembly experiments, as one includes additional fully connected second layers, the total number of parameters is $n_l \times l^2/s^2 \times n_f + K_s^2 \times n_c \times n_f + n_f$, where $n_l$ is the number of labels (here, $n_l = 10$).

### B.3 INFERENCE WITH THE IMPLICIT GUMBEL-SINKHORN DISTRIBUTION

Here we show how to compute $KL((X + \varepsilon)/\tau \parallel \varepsilon/\tau_{prior})$, as defined in 4.1. We first notice that the density of the variable $h = (a + g)/b$, where $g$ has a Gumbel distribution and $a, b$ are constants is given by:

$$\log p_h(z) = \log b - (bz - a + \exp(a - bz)). \tag{15}$$

Therefore, the log density ratio $LR(z)$ between each component of $h_1 = (x_{i,j} + \varepsilon_{i,j})/\tau$ and $h_2 = \varepsilon_{i,j}/\tau_{prior}$ is (suppressing indexing for simplicity)

$$LR(z) = \log p_{h_1}(z)/\log p_{h_2}(z)$$
$$= \log \tau - (\tau z - x + \exp(x - z\tau)) - \log \tau_{prior} + (\tau_{prior}z + \exp(-z\tau_{prior})).$$

We need to take expectations with respect to the distribution of $h_1$. To compute this expectation, we first express the above ratio in terms of $\varepsilon$

$$LR(\varepsilon) = \log(\tau/\tau_{prior}) - (\varepsilon + \exp(-\varepsilon) - (\varepsilon + x)\tau_{prior}/\tau - \exp(-(\varepsilon + x)\tau_{prior}/\tau)))$$

Now we appeal to the *law of the unconscious statistician*, and take the expectation with respect to $\varepsilon$. Using the identities

- $E(\varepsilon) = \gamma \approx 0.5772$ (the Euler-Mascheroni constant)
- Moment generating function $E(\exp(t\varepsilon)) = \Gamma(1 - t)$; implying $E(\exp(-\varepsilon)) = 1$ and $E(\exp(-\tau_{prior}/\tau\varepsilon)) = \Gamma(1 + \tau_{prior}/\tau))$

we have:

$$E_{h_1}(LR(z)) = E_\varepsilon(LR(\varepsilon))$$
$$= \log(\tau/\tau_{prior}) - (\gamma(1 - \tau_{prior}/\tau) + 1 - x\tau_{prior}/\tau - \exp(-x\tau_{prior}/\tau)\Gamma(1 + \tau_{prior}/\tau)).$$

From this, it easily follows (adding all the $N^2$ components) that

$$KL((X + \varepsilon)/\tau \parallel \varepsilon/\tau_{prior}) = \sum_{i,j} E_{g_1}(LR(z_{i,j}))$$
$$= N^2 (\log(\tau/\tau_{prior}) - 1 + \gamma(\tau_{prior}/\tau - 1)) + S_1 + \Gamma(1 + \tau_{prior}/\tau)S_2,$$

where $S_1 = \tau_{prior}/\tau \sum_{i,j} x_{i,j}$ and $S_2 = \sum_{i,j} \exp(-x_{i,j}\tau_{prior}/\tau)$.

## C SUPPLEMENTAL RESULTS

### C.1 PUZZLES

In table 4 we provide further performance measures for the Jigsaw puzzle task on Celeba, for extreme hyper-parameter values: small temperature, large temperature, and a single Sinkhorn iteration These are worse than the ones in table 2, although surprisingly, one Sinkhorn iteration already provides reasonable performance, as long temperature is chosen in an appropriate range.

### C.2 TRANSFORMATIONS INTO ARBITRARY DIGITS

In table 5 we show performance of a 2-layer CNN in detecting transformed digits as the ones they are intended to be. From this we see the most troublesome transformation was to one, as this network most of the times categorized it as a different number. Also, in figure 4 we show transformations, showing that to reconstruct to arbitrary digits it is not required that the original ones have an actual digit-like structure, but they can be only pieces of 'strokes' or 'dust'.

Table 4: Jigsaw puzzle results for different extreme hyper-parameter values

| | $\tau = 0.01$ | | | | $\tau = 100$ | | | | $L = 1$ | | | |
|---|---|---|---|---|---|---|---|---|---|---|---|---|
| | 2x2 | 3x3 | 4x4 | 5x5 | 2x2 | 3x3 | 4x4 | 5x5 | .2x2 | 3x3 | 4x4 | 5x5 |
| Prop. wrong | .06 | .08 | .23 | .36 | .03 | .1 | .28 | .5 | .0 | .03 | .13 | .28 |
| Prop. any wrong | .1 | .22 | .36 | .9 | .04 | .23 | .67 | .97 | .0 | .08 | .42 | .82 |
| Kendall tau | .9 | .89 | .74 | .62 | .97 | .88 | .7 | .47 | 1.0 | .96 | .86 | .72 |
| $l1$ | .03 | .04 | .1 | .14 | .01 | .04 | .11 | .19 | .0 | .01 | .05 | .11 |
| $l2$ | .16 | .18 | .28 | .34 | .11 | .19 | .3 | .38 | .0 | .11 | .21 | .3 |

| | | | | | | Becomes | | | | | |
|---|---|---|---|---|---|---|---|---|---|---|---|
| | | 0 | 1 | 2 | 3 | 4 | 5 | 6 | 7 | 8 | 9 |
| | 0 | 1. | .0 | 1. | 1. | 1. | 1. | 1. | 1. | 1. | 1. |
| | 1 | .91 | 1. | .97 | .99 | .99 | 1. | 1. | .56 | .75 | .2 |
| | 2 | 1. | .0 | 1. | 1. | 1. | 1. | 1. | .70 | 1. | 1. |
| Actual digit | 3 | .04 | .0 | 1. | 1. | 1. | 1. | .96 | 1. | 1. | .96 |
| | 4 | 1. | .46 | 1. | 1. | 1. | 1. | 1. | 1. | .68 | .36 |
| | 5 | 1. | .0 | 1. | 1. | .63 | 1. | 1. | 1. | 1. | 1. |
| | 6 | .3 | .01 | 1. | 1. | 1. | 1. | .65 | 1. | .65 | 1. |
| | 7 | .0 | .73 | .27 | .46 | 1. | 1. | 1. | 1. | 1. | .72 |
| | 8 | 1. | .07 | 1. | 1. | 1. | 1. | 1. | .07 | 1. | 1. |
| | 9 | 1. | .33 | 1. | 1. | 1. | 1. | 1. | 1. | 1. | .66 |

Table 5: Accuracies of two-layer convolutional neural network in identifying transformed digits

## C.3 RESULTS ON CATEGORIAL VAE IN MNIST

In general, for arbitrary random variables $Z_1, Z_2$ and a function $g$, one has

$$KL(Z_1 \parallel Z_2) \geq KL(g(Z_1) \parallel g(Z_2)). \tag{16}$$

We prove this in the discrete case, for simplicity: call $q(z)$ and $p(z)$ the densities of $Z_1, Z_2$, and call $y = g(z)$. This induces two joint distributions, $p(z, y)$ and $q(z, y)$. Now, define

$$KL(q(z|y) \parallel p(z|y)) = \sum_{y,z} (q(z, y) \log q(z|y) - \log p(z|y)).$$

Under this definition, one can verify that

$$
\begin{aligned}
KL(q(z, y) \parallel p(z, y)) =& KL(q(z) \parallel p(z)) + KL(q(y|z) \parallel p(y|z)) \\
=& KL(q(y) \parallel p(y)) + KL(q(z|y) \parallel p(z|y)).
\end{aligned}
$$

But $KL((q(y|z) \parallel p(y|z)) = 0$, as $y$ is a deterministic function of $z$. Therefore, $KL((q(z) \parallel p(z)) = KL(q(y) \parallel p(y)) + KL(q(z|y) \parallel p(z|y))$, and since the second term is positive (a KL divergence) we conclude $KL(q(z) \parallel p(z)) \geq KL(q(y) \parallel p(y))$.

This implies a lower (or less tight) ELBO if using $Z_1, Z_2$ instead of $g(Z_1), g(Z_2)$. However, we note that in the categorical case this has a minimal impact in performance. Indeed, we replicated the density estimation on MNIST task described in Jang et al. (2016); Maddison et al. (2016), and as alternative method we considered the concrete distribution, but using as stochastic node $(\varepsilon + x)/\tau$ (with prior $\varepsilon/\tau_{prior}$ instead of two concrete distributions. In other words, for us $g(x) = \text{softmax}_\tau(x)$ and $Z_1 = (\varepsilon + x)/\tau, Z_2 = (\varepsilon)/\tau_{prior}$ (in law). Results are shown in Table 6. We first see that Concrete distribution does worse than Gumbel-Softmax, which we attribute to a sub-optimal parameter search. However, we see that working in the Gumbel space has little impact on $\log p(x)$: the difference was smaller than .5 nats.

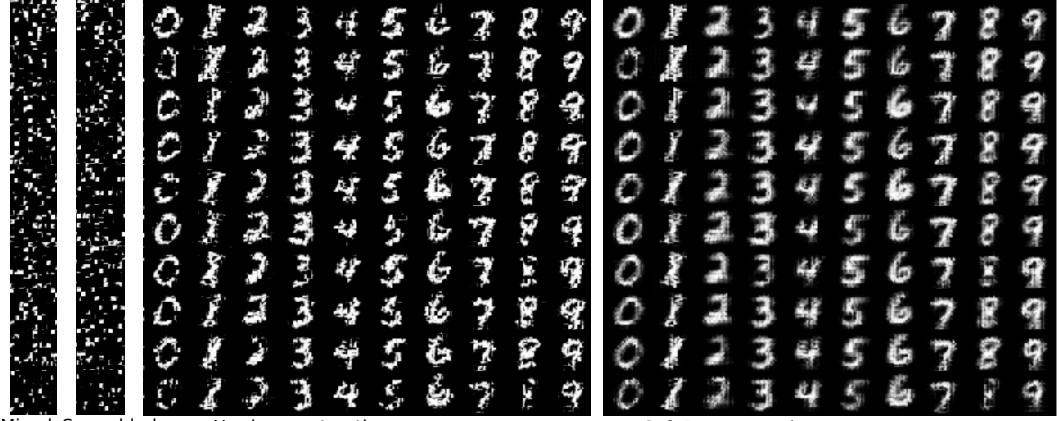

Mixed  Scrambled        Hard reconstructions                    Soft Reconstructions

Figure 4: First column: samples from dataset created by mixing all pieces of digits, and then re-assembling them into 'digits'. Second column: random permutations of first column. Third column: hard reconstructions using $M(X)$. Fourth column: soft reconstructions using $S(X/\tau)$ and $\tau = 1$. Metaphorically, one is able to reconstruct pieces out of 'dust'.

| Method | $-\log p(x)$ |
|---|---|
| Gumbel-Softmax | 106.7 |
| Concrete | 111.5 |
| Concrete (Gumbel space) | 111.9 |

Table 6: Summary of results in VAE

| Mean number of candidates | 10 | | 30 | | 45 | | 60 | |
|---|---|---|---|---|---|---|---|---|
| Difficulty | 1 worm | 4 worms | 1 Worm | 4 worms | 1 worm | 4 worms | 1 worms | 4 worms |
| MCMC | .34 | .65 | .18 | .28 | .14 | .17 | .13 | .16 |
| (Linderman et al., 2017) | .77 | .93 | .33 | **.7** | .18 | .48 | .17 | .37 |
| Gumbel-Sinkhorn | **.79** | **.94** | **.4** | .69 | **.25** | **.51** | **.21** | **.44** |
| Gumbel-Sinkhorn (no regularization) | 0.77 | .92 | **.4** | .64 | .25 | .44 | .21 | .39 |

Table 7: Accuracy in the C.elegans neural identification problem, for varying mean number of candidate neurons (10, 30, 45, 60) and number of worms (1 and 4).

### C.4 Supplementary results on C.elegans

Finally, in Table 7 we show additional results for the C.elegans experiment. The setting is the same as in Figure 4(a) in Linderman et al. (2017). Likewise, Table 3 correspond to the setting of Figure 4(b) in Linderman et al. (2017).

## D Supplementary discussion

### D.1 Sinkhorn operator for approximate marginal inference

A second connection between the distribution in (6) (and therefore, the Matching Gumbel distribution) and the Sinkhorn operator arises as a consequence of Theorem 1. This relates to the estimation of the marginals $E_\theta(P_{i,j})$, known to be a #P hard problem. A well known result (Globerson & Jaakkola, 2007; Wainwright et al., 2008), consequence of Fenchel (conjugate) duality (Rockafellar, 1970) applied to exponential families, links this problem to optimization in the following way: lets denote by $\mathcal{M}$ the marginal polytope, the convex hull of the set of realizable sufficient statistics, that here coincides with $\mathcal{B}_n$. Also, lets call $\mathcal{H}(\mu)$ the entropy of (6) for the parameter $\theta(\mu)$ such that

$\mu = E_{\theta(\mu)}(P)$. Then,

$$E_\theta(P) = \arg\max_{\mu \in \mathcal{M}} \langle \theta, \mu \rangle_F + \mathcal{H}(\mu). \tag{17}$$

Notice the only difference between the optimization problems in (17) and (10) is the entropy term, after identifying $X$ with $\theta$. Therefore, one may understand the Sinkhorn operator as providing approximations for the partition function and the marginals, which will be accurate insofar as $h(\mu)$ is a good approximation for $\mathcal{H}(\mu)$. In this way, one can understand $S(X)$ as an approximation for $E_\theta(P)$, that may complement more classical ones, as the Bethe and Kituchani's approximations for $\mathcal{H}(\mu)$, and the corresponding approximate inference algorithms that they give rise to (Yedidia et al., 2001; Vilnis et al., 2015).

## D.2 SUMMARY OF EXTENSIONS

Table 8: Analogies between permutation and categories

| | **Categories** | **Permutations** |
|---|---|---|
| **Polytope** | Probability simplex $\mathcal{S}$ | Birkhoff polytope $\mathcal{B}_\mathcal{N}$ |
| **Linear program** | $\arg\max x_i = \arg\max_{s \in \mathcal{S}} \langle x, s \rangle$ | $M(X) = \arg\max_{P \in \mathcal{B}} \langle P, X \rangle_F$ |
| **Approximation** | $\arg\max_i x_i = \lim_{\tau \to 0^+} \text{softmax}(x/\tau)$ | $M(X) = \lim_{\tau \to 0^+} S(X/\tau)$ |
| **Entropy** | $h(s) = \sum_i -s_i \log s_i$ | $h(P) = \sum_{i,j} -P_{i,j} \log(P_{i,j})$ |
| **Entropy regularized linear program** | $\text{softmax}(x/\tau) = \arg\max_{s \in \mathcal{S}} \langle x, s \rangle + \tau h(s)$ | $S(X/\tau) = \arg\max_{P \in \mathcal{B}} \langle P, X \rangle_F + \tau h(P)$ |
| **Reparameterization** | **Gumbel-max trick** $\arg\max_i(x_i + \epsilon_i)$ | **Gumbel-Matching** $\mathcal{G}M(X)$ $M(X + \epsilon)$ |
| **Continuous approximation** | **Concrete** $\text{softmax}((x + \epsilon)/\tau)$ | **Gumbel-Sinkhorn** $\mathcal{G}S(X, \tau)$ $S((X + \epsilon)/\tau)$ |

