# OpenReview forum: "Learning Latent Permutations with Gumbel-Sinkhorn Networks"
_ICLR.cc/2018/Conference — Accept (Poster)_

### Official Review · AnonReviewer3 · 2017-11-26
**The paper utilizes finite approximation of the Sinkhorn operator to describe how one can construct a neural network for learning from permutation valued training data. A probabilistic view of permutation via injection of Gumbel noise is also presented in the paper.**

**Rating:** 8
**Confidence:** 4

**Review:**

Quality: The paper is built on solid theoretical grounds and supplemented by experimental demonstrations. Specifically, the justification for using the Sinkhorn operator is given by theorem 1 with proof given in the appendix. Because the theoretical limit is unachievable, the authors propose to truncate the Sinkhorn operator at level $L$. The effect of approximation for the truncation level $L$ as well as the effect of temperature $\tau$ are demonstrated nicely through figures 1 and 2(a). The paper also presents a nice probabilistic approach to permutation learning, where the doubly stochastic matrix arises from Gumbel matching distribution.

Clarity: The paper has a good flow, starting out with the theoretical foundation, description of how to construct the network, followed by the probabilistic formulation. However, I found some of the notation used to be a bit confusing.

1. The notation $l$ appears in Section 2 to denote the number of iterations of Sinkhorn operator. In Section 3, the notation $l$ appears as $g_l$, where in this case, it refers to the layers in the neural network. This led me to believe that there is one Sinkhorn operator for each layer of neural network. But after reading the paper a few times, it seemed to me that the Sinkhorn operator is used only at the end, just before the final output step (the part where it says the truncation level was set to $L=20$ for all of the experiments confirmed this). If I'm correct in my understanding, perhaps different notation need to be used for the layers in the NN and the Sinkhorn operator. Additionally, it would have been nice to see a figure of the entire network architecture, at least for one of the applications considered in the paper.

2. The distinction between $g$ and $g_l$ was also a bit unclear. Because the input to $M$ (and $S$) is a square matrix, the function $g$ seems to be carrying out the task of preparing the final output of the neural network into the input formate accepted by the Sinkhorn operator. However, $g$ is stated as "the output of the computations involving $g_l$". I found this statement to be a bit unclear and did not really describe what $g$ does; of course my understanding may be incorrect so a clarification on this statement would be helpful.

Originality: I think there is enough novelty to warrant publication.  The paper does build on a set of previous works, in particular Sinkhorn operator, which achieves continuous relaxation for permutation valued variables. However, the paper proposes how this operator can be used with standard neural network architectures for learning permutation valued latent variable. The probabilistic approach also seems novel. The applications are interesting, in particular, it is always nice to see a machine learning method applied to a unique application; in this case from computational neuroscience.

Other comments:

1. What are the differences between this paper and the paper by Adams and Zemel (2011)? Adams and Zemel also seems to propose Sinkhorn operator for neural network. Although they focus only on the document ranking problem, it would be good to hear the authors' view on what differentiates their work from Adams and Zemel.

2. As pointed out in the paper, there is a concurrent work: DeepPermNet. Few comments regarding the difference between their work and this work would also be helpful as well.

Significance: The Sinkhorn network proposed in the paper is useful as demonstrated in the experiments. The methodology appears to be straight forward  to implement using the existing software libraries, which should help increase its usability.

The significance of the paper can greatly improve if the methodology is applied to other popular machine learning applications such as document ranking, image matching, DNA sequence alignment, and etc. I wonder how difficult it is to extend this methodology to bipartite matching problem with uneven number of objects in each partition, which is the case for document ranking. And for problems such as image matching (e.g., matching landmark points), where each point is associated with a feature (e.g., SIFT), how would one formulate such problem in this setting?

---

> ### Author Response · Authors · 2018-01-03
> **Official Response**
>
> We thank the reviewer for the good evaluation of our paper, and the useful commentary.
>
> 1) We unintentionally overloaded the $l$ index, and we will fix this final paper.  The Sinkhorn operator is only applied to the output of the last layer of the neural network.  We will include a new figure (in the appendix) depicting our architecture.
>
> 2) We also agree the notation with $g$ and $g_l$ currently is odd. We will make an attempt to improve exposition in the main text, by defining g as the composition of g_l. The new figure that depicts the architecture should also help.
>
> 3) Our work drew some inspiration from Adams and Zemel, but it has clear differences:  i) we consider different tasks, beyond ranking; ii) we use a shared architecture to save parameters (permutation equivariance); and iii) in Adams and Zemel the Sinkhorn operator is used to heuristically approximate an expectation over permutations. Theorem 1 allows us to justify that approximation: by appealing to the framework of Variational Inference in exponential families [1] we can understand the Sinkhorn operator as an approximate marginal inference routine. We will briefly comment on that in the related work section and as a new last appendix (see also response to AnonReviewer2, point 1). Also, we will improve this section in order to make these distinctions more clear.
>
> 4) DeepPermNet obtains results that are comparable to ours; however, our architecture is much simpler, as argued in the results section and appendix B.2.
>
> 5) We agree that there are many ways in which this work could be extended, and we are actively investigating some of them. In cases of matchings between groups of different sizes, there is a simple extension: pad the cost matrix with zeros so that its row and column dimensions coincide. Also, regarding image matchings, they may be achieved by changing the architecture slightly: this is indeed explored in a simultaneous ICLR submission [2], dealing with generative models from an optimal transportation perspective. There, a network is trained to match samples from two datasets (the actual data and samples from the generative model) so that minimizes a total cost functional that is the distance on some learned embedding space.
>
>
> [1] Graphical Models, Exponential Families, and Variational Inference Martin J. Wainwright. and Michael I. Jordan. https://people.eecs.berkeley.edu/~wainwrig/Papers/WaiJor08_FTML.pdf
> [2]Improving GANs Using Optimal Transport. https://openreview.net/forum?id=rkQkBnJAb

---

### Official Review · AnonReviewer1 · 2017-11-27
**Gumbel-Sinkhorn networks for learning permutations**

**Rating:** 7
**Confidence:** 4

**Review:**

Learning latent permutations or matchings is inherently difficult because the marginalization and partition function computation problems at its core are intractable. The authors propose a new method that approximates the discrete max-weight matching by a continuous Sinkhorn operator, which looks like an analog of softmax operator on matrices. They extend the Gumbel softmax method (Jang et al., Maddison et al. 2016) to define a Gumbel-Sinkhorn method for distributions over latent matchings. Their empirical study shows that this method outperforms competitive baselines for tasks such as sorting numbers, solving jigsaw puzzles etc.

In Theorem 1, the authors show that Sinkhorn operator solves a certain entropy-regularized problem over the Birkhoff polytope (doubly stochastic matrices). As the regularization parameter or temperature \tau tends to zero, the continuous solution approaches the desired best matching or permutation. An immediate question is, can one show a convergence bound to determine a reasonable choice of \tau?

The authors use the Gumbel trick that recasts a difficult sampling problem as an easier optimization problem. To get around non-differentiable re-parametrization under the Gumbel trick, they extend the Gumbel softmax distribution idea (Jang et al., Maddison et al. 2016) and consider Gumbel-Sinkhorn distributions. They illustrate that at low temperature \tau, Gumbel-matching and Gumbel-Sinkhorn distributions are indistinguishable. This is still not sufficient as Gumbel-matching and Gumbel-Sinkhorn distributions have intractable densities. The authors address this with variational inference (Blei et al., 2017) as discussed in detail in Section 5.4.

The empirical results do well against competitive baselines. They significantly outperform Vinyals et al. 2015 by sorting up to N = 120 uniform random numbers in [0, 1] with great accuracy < 0.01, as opposed to Vinyals et al. who used a more complex recurrent neural network even for N = 15 and accuracy 0.9.

The empirical study on jigsaw puzzles over MNIST, Celeba, Imagenet gives good results on Kendall tau, l1 and l2 losses, is slightly better than Cruz et al. (arxiv 2017) for Kendall tau on Imagenet 3x3 but does not have a significant literature to compare against. I hope the other reviewers point out references that could make this comparison more complete and meaningful.

The third empirical study on the C. elegans neural inference problem shows significant improvement over Linderman et al. (arxiv 2017).

Overall, I feel the main idea and the experiments (especially, the sorting and C. elegance neural inference) merit acceptance. I am not an expert in this line of research, so I hope other reviewers can more thoroughly examine the heuristics discussed by the authors in Section 5.4 and Appendix C.3 to get around the intractable sub-problems in their approach.

---

> ### Author Response · Authors · 2018-01-03
> **Official response**
>
> We thank the reviewer for their good evaluation of our paper, and the useful commentary.
>
> 1) The question of convergence bounds is a quite relevant one, and we stress there is a double limit, involving tau and L. A rigorous analysis of such convergence goes beyond the scope of our work, but we point to recent convergence bounds results [1] in the more general entropy regularized OT problem, that in our case expresses in terms of optimization over the Birkhoff polytope. We believe research as in [1] is highly relevant to our work, as it suggests ways to obtain computational improvements by suitably tweaking the plain Sinkhorn iteration scheme. We plan to explore this research avenue in the future.  For now, choice of tau is treated as a hyperparameter, we select it so that performance is optimal. This is discussed in the main text (section 3) and the appendix C.1
>
> 2) We did not include more results related to jigsaw solving with neural networks as this problem is very recent in the context of neural networks. Nonetheless, we include a reference to another paper [2] that deals with jigsaw puzzles using neural networks, although comparisons are impossible since they work on a i) different dataset (Pascal VOC) and ii)their method does not scale with permutations, as it does not appeal to Sinkhorn operator but indexes each permutation as a separate entity, and limits the number of used permutations to at most 1000.
>
>
> [1]Near-linear time approximation algorithms for optimal transport via Sinkhorn iteration Jason Altschuler∗ , Jonathan Weed∗ , and Philippe Rigollet. https://arxiv.org/pdf/1705.09634.pdf
>  The concrete distribution. https://arxiv.org/abs/1611.00712
> [2]M. Noroozi and P. Favaro. Unsupervised learning of visual representations by solving jigsaw puzzles.https://arxiv.org/abs/1603.09246

---

### Official Review · AnonReviewer2 · 2017-11-30
**An interesting paper, based on a simple and neat idea, with good experimental results**

**Rating:** 6
**Confidence:** 2

**Review:**

The idea on which the paper is based - that the limit of the entropic regularisation over Birkhoff polytope is on the vertices = permutation matrices -, and the link with optimal transport, is very interesting. The core of the paper, Section 3, is interesting and represents a valuable contribution.

I am wondering whether the paper's approach and its Theorem 1 can be extended to other regularised versions of the optimal transport cost, such as this family (Tsallis) that generalises the entropic one:

https://aaai.org/ocs/index.php/AAAI/AAAI17/paper/view/14584/14420

Also, it would be good to keep in mind the actual proportion of errors that would make a random choice of a permutation matrix for your Jigsaws. When you look at your numbers, the expected proportion of parts wrong for a random assignment could be competitive with your results on the smallest puzzles (typically, 2x2). Perhaps you can put the *difference* between your result and the expected result of a random permutation; this will give a better understanding of what you gain from the non-informative baseline.
(also, it would be good to define "Prop. wrong" and "Prop. any wrong". I think I got it but it is better to be written down)

There should also be better metrics for bigger jigsaws -- for example, I would accept bigger errors if pieces that are close in the solution tend also to be put close in the err'ed solution.

Typos:

* Rewrite definition 2 in appendix. Some notations do not really make sense.

---

> ### Author Response · Authors · 2018-01-04
> **Official Response**
>
> We thank the reviewer for their thorough evaluation and thoughtful comments..
>
> 1) Regarding Tsallis entropy, we recognize it is a quite interesting direction, as it allows to better understand our work in terms of information geometry (e.g. [1]). In a revised version we will include a commentary on how Theorem 1 may be interpreted  as a way of performing marginal in exponential families (e.g. see [2]). With this, it will become clear using Tsallis entropy would yield yet another approximation. However, it is not clear that increases in computational complexity would justify the use of this type of entropy. See also response to AnonReviewer 3 for a complementary discussion.
>
> 2) By “proportion wrong” we mean the proportion of wrongly identified pieces. By “proportion any wrong” we consider the proportion of cases (entire puzzles) where there was at least one mistake. The latter was used as a performance measure in [3], and it is much more stringent. 	We will clarify this in the main text.
>
> 3)Regarding the proportion wrong: the expected value of the proportion of errors in random guessing a permutation of n items  can be shown to be (n-1)/(n). In 2x2 puzzles, this means we expect 75% of wrong pieces at random, but in practice no errors occur with our method (that is an easy case, though). We will comment on this baseline in a revised version
>
> 4) We agree there might be better ways to measure error. For example, if we shift one row by one and put the last element at first, then the error on that row will be 100%. It will be also high according to the  l1 and l2 norms. However, the solution still makes sense from the point of view of preserving locality constraints and e.g. still look good on vision tasks. Because of this we believe we should move towards using metrics that take into consideration the structure of local coherence between pieces. Unfortunately, that goes beyond the scope of this work.
>
>  5) We will correct the typo and minor inconsistencies in the main text and appendix.
>
> We hope the reviewer will take the reviews with higher degree of confidence into consideration.
>
> [1]S.I. Amari. Information geometry and its applications http://www.springer.com/gp/book/9784431559771
> [2] Graphical Models, Exponential Families, and Variational Inference Martin J. Wainwright. and Michael I. Jordan. https://people.eecs.berkeley.edu/~wainwrig/Papers/WaiJor08_FTML.pdf
> [3] Order Matters: Sequence to sequence for sets. Oriol Vinyals, Samy Bengio, Manjunath Kudlur]https://arxiv.org/abs/1511.06391

---

### Public Comment · ~Patrick_Emami1 · 2017-12-03
**Small typo in Appendix B.2**

Thank you for this excellent work!

If I am not mistaken, I believe there is a small typo in Appendix B.2. The number of parameters of the simple network architecture should be "n_u + N x n_u", not "N + N x n_u". Since the hidden layer has n_u units, you need n_u parameters to map each number to an n_u-dimensional space. Then, it takes N x n_u parameters to produce each N-dimensional row of g(X; theta)?

---

> ### Author Response · Authors · 2017-12-14
> **Thanks**
>
> Dear Patrick,
>
> Thanks for the kind words about our work. We are thankful that you noticed our mistake. It has been amended in the revised version we are soon to submit. We also extrapolated your remark and checked for consistency in the remaining formulae at appendix B2.
> The authors

---

### Author Response · Authors · 2018-01-04
**Rebuttal/Revised Version**

Dear community, we have uploaded a revised version of our manuscript. Our changes are summarized as follows:

1) The reviewers provided very useful criticism, suggesting directions for improvement and detecting current minor weaknesses/inconsistencies. We have implemented their suggestions. Our responses to individual reviewers contain those changes. As a result, now we can present a stronger paper.
2) We corrected other typos and/or notational inconsistencies.
3) To improve the flow of the narrative, we changed the current exposition of the approximate posterior inference method, by moving it from the appendix B.2 and section 5.4 back to section 4.
4) We released code for the Gumbel-Sinkhorn estimator, applied to the number sorting problem. This actual link will be included in the final version, to preserve anonymity.
5) We expanded our Related Work section to discuss our work in the light of recent literature; notably a concurrent ICLR submission”Improving GANs Using Optimal Transport
 https://openreview.net/forum?id=rkQkBnJAb and Learning Generative Models with Sinkhorn Divergences https://arxiv.org/abs/1706.00292
6)We included a new Figure (new Figure 1), depicting the neural network architecture. Previous Figure 1 was moved to the appendix, as Figure 3

---

### Decision · Program_Chairs · 2018-01-29
**ICLR 2018 Conference Acceptance Decision**

**Decision:**

Accept (Poster)

**Comment:**

This paper with the self-explanatory title was well received by the reviewers and, additionally, comes with available code. The paper builds on prior work (Sinkhorn operator) but shows additional, significant amount of work to enable its application and inference in neural networks.  There were no major criticisms by the reviewers, other than obvious directions for improvement which should have been already incorporated in the paper, issues with clarity and a little more experimentation. To some extent, the authors addressed the issues in the revised version.